# Switching from Rasagiline to Safinamide as an Add-On Therapy Regimen in Patients with Levodopa: A Literature Review

**DOI:** 10.3390/brainsci13020276

**Published:** 2023-02-07

**Authors:** Pilar Sanchez Alonso, Beatriz De La Casa-Fages, Araceli Alonso-Cánovas, Juan Carlos Martínez-Castrillo

**Affiliations:** 1Movement Disorders Unit, Neurology Department, Hospital Universitario Puerta de Hierro Majadahonda, 28222 Madrid, Spain; 2Movement Disorder Unit, Hospital General Universitario Gregorio Marañón, 28007 Madrid, Spain; 3Movement Disorders Unit, Neurology Department, Hospital Universitario Ramón y Cajal, 28034 Madrid, Spain

**Keywords:** Parkinson’s disease, MAO-B inhibitors, safinamide, rasagiline, adjunctive therapy, therapy switch

## Abstract

Parkinson’s disease (PD) is a complex disease, and the treatment is focused on the patient’s clinical symptoms. Levodopa continues to be the most effective drug for symptomatic PD treatment. However, chronic levodopa treatment is associated with the development of motor complications in most patients. Add-on therapeutic drugs, such as dopamine agonists and monoamine oxidase B (MAO-B) inhibitors, for example, safinamide and rasagiline, may be a desirable addition to continuously increase the levodopa dose for the optimization of motor control in PD. The scientific literature shows that safinamide significantly alleviated motor fluctuations with no increase in troublesome dyskinesia, thanks to its unique double mechanism, providing further benefits to fluctuating PD patients when compared to a placebo or other drugs. Switching from rasagiline to safinamide has been shown to improve the wearing-off phenomena, which is defined as the recurrent, predictable worsening of symptoms of parkinsonism at the end of the levodopa dose until the next dose reaches a clinical effect. In this situation, safinamide may be helpful for reducing the total daily dose of levodopa, improving the OFF time and ON time without troublesome dyskinesias, and being more effective than other MAO-B inhibitors. In this narrative review, we explore the switch from rasagiline to safinamide in patients with motor complications as a feasible and effective alternative to optimize antiparkinsonian treatment.

## 1. Introduction

Parkinson’s disease (PD) is a complex, progressive, and age-related condition characterized by the loss of dopaminergic neurons in the pars compacta of the substantia nigra, as well as the accumulation of misfolded α-synuclein called Lewy bodies [1,2,3]. It occurs due to the extensive damage of dopamine-producing neurons, which, in turn, causes dopamine deficits in the midbrain, followed by the alternation of various other neurotransmitters (glutamate, GABA, serotonin, etc.) [4]. It has been observed that the fluctuation of neurotransmission in the basal ganglia exhibits a great impact on the pathophysiology of PD [4]. The neuropathology of PD has shown that complex, interconnected neuronal systems, regulated by several neurotransmitters in addition to dopamine, are involved in the etiology of motor and non-motor symptoms [5].

Levodopa continues to be the most effective drug for symptomatic PD treatment. Chronic levodopa treatment, however, is associated with motor complications in most patients as their disease progresses. Therefore, it is recommended that clinicians initiate levodopa treatment with low doses in PD patients and gradually increase it, using the lowest dose that provides satisfactory clinical control, especially in younger patients and women, who are more likely to develop dyskinesia [6]. A suggested threshold of 400 mg/day is recommended, provided that the clinical requirements of dopaminergic therapy are met [3,6,7]. Some authors reported that the add-on drugs, such as dopamine agonists and monoamine oxidase B (MAO-B) inhibitors, may be a preferable option to continuously increase the levodopa dose [6,8,9]. An MAO-B inhibitor can be used for this purpose in patients with PD experiencing motor fluctuations. Inhibiting MAO-B-mediated dopamine metabolism is key to the therapeutic efficacy of these drugs [6]. Although rasagiline and safinamide are two members of the MAO-B inhibitors, there are differences between them [9,10,11]. Rasagiline can either be used as a monotherapy (without levodopa) or as an adjunct therapy (with either levodopa or dopamine agonists) for PD. In contrast, safinamide is used to treat Parkinson’s patients with fluctuating mid-to-late-stage symptoms along with a stable dose of levodopa, either alone or in combination with other treatments [9,10]. Numerous studies have shown that MAO-B inhibitors, such as safinamide and rasagiline, are effective and safe in combination with levodopa [11,12,13]. Safinamide (Xadago^®^, Zambon S.p.A., Bresso, Italy) is an orally administered α-aminoamide derivative, which combines the strong, selective, and reversible inhibition of monoamino oxidase B (MAO-B) through a blockade of voltage-dependent Na+ and Ca2+ channels and the inhibition of stimulated glutamate release [12]. Specifically, it targets both dopaminergic and glutamatergic receptors (see Figure 1) [13].

Among the most common adverse effects reported in clinical trials with safinamide 100 mg are dyskinesia, falls, and nausea. The drug has also been associated with less frequent adverse effects such as hypertension, indigestion, sleepiness, and drowsiness [13]. There is a potential risk of hypertensive crisis when Safinamide is combined with other MAOIs [13]. It is also contraindicated to use safinamide with opioids [13]. Rasagiline is likely to exert its primary effect on PD by inhibiting MAO-B, thereby slowing down the metabolism of endogenous and exogenous dopamine [15]. Headaches, dizziness, and insomnia are among the most frequently reported AEs associated with rasagiline [16]. It has been shown in a systematic review that rasagiline was not associated with more symptoms than the placebo in trials [16].

Rasagiline inhibits MAO-B irreversibly, causing striatal extracellular dopamine levels to increase. This increase in the concentration of dopamine and dopaminergic activity is the likely mediator of the beneficial effects of rasagiline, as observed in animal models with dopaminergic motor dysfunction [17]. The only reversible MAO-B inhibitor that has demonstrated greater bioavailability and selectivity than other MAO-B inhibitors is safinamide [14]. Due to the additional activity of glutamate release modulation, its metabolites are inert, adverse events are moderate, and motor fluctuations are improved along with non-motor symptoms in the advanced stage [15,16,17]. A 50 mg or 100 mg dose regimen is currently available for safinamide. Clinically, both doses are equivalent to 1 mg of rasagiline for LD-equivalent daily dose (LEDD) calculations [18,19,20]. In addition to its indirect dopamine-enhancing effect, high doses of safinamide (100 mg) also block voltage-gated Na+ and Ca++ channels and inhibits glutamate release at overactive synapses [6,20,21]. In untreated PD patients, glutamate hyperexcitability is restricted to cortical motor areas, and functionally related basal ganglia regions emerge early. Fluctuations and dyskinesias are correlated with postsynaptic striatal changes in glutamatergic receptor density and sensitivity [22,23]. It is well known that excessive glutamate concentrations or the hyperactivity of glutamatergic nerve terminals can lead to neuronal damage and may be involved in the neurodegeneration associated with PD [23,24].

Clinical trials have shown that safinamide can improve fluctuations and treat OFF periods when switched from rasagiline [18]. To prevent serotonin syndrome and hypertension crisis, it is recommended that physicians follow a 15-day suspension period between MAO-B inhibitors when switching between them [18].

The use of MAO-B inhibitors as a treatment for Parkinson’s disease has demonstrated therapeutic advantages, good tolerability, and safety, as well as a low incidence of adverse events. We aim to review the current state of knowledge regarding the therapeutic use of safinamide and rasagiline in PD patients as well as compile all evidence regarding switching between rasagiline and safinamide.

## 2. Materials and Methods

We searched the following databases for published studies in English and Spanish without time restrictions: Medline (via PubMed) and the Cochrane central register of controlled trials (CENTRAL). The following terms were used to generate a search: “Parkinson* Disease”, “Parkinson* disorders”, “Monoamine oxidase* inhibitor*”, “Safinamide”, and “Rasagiline”. As per the aim of this study, we included only studies performed on humans. All preclinical studies, in vitro studies, or studies that included animals were excluded. Additionally, the grey literature was examined via Google Scholar. In order to identify additional relevant citations, we examined the reference lists and citation indices of the articles included in the study.

## 3. Results

### 3.1. Evidence of Efficacy of Safinamide and Rasagiline on Motor Symptoms

The efficacy and safety of safinamide (50 mg/day and 100 mg/day) as an add-on therapy to levodopa in patients with mid-to-late stage Parkinson’s disease with motor fluctuations have been examined in two randomized, double-blind, placebo-controlled trials [19,20], and one extension study [21] (See Table 1). Based on the results of Study 016, both doses of safinamide improved the total on-time at week 24 with no or non-troublesome dyskinesia (a statistically significant difference of 0.55 h; 95% CI, 0.12–0.99; *p* = 0.0130) and off-time (the average time with safinamide was about 1.3 h and the average time with a placebo). Moreover, it also showed improvements in motor function (change in UPDRS-III from baseline) and the patient’s overall clinical status and activities of daily living [19]. A follow-up study was approved for an additional 18 months for most of the patients who completed this trial. Ad hoc subgroup analysis of moderate to severe dyskinetic patients showed a reduction with safinamide 100 mg/d compared with the placebo, although the “Change in Dyskinesia Rating Scale” was not significantly different between the two groups. After six months, improvements in motor function, activities of daily living, depressive symptoms, clinical status, and quality of life persisted for 24 months [21].

The phase III SETTLE study was a 6-month (24-week) randomized, double-blind, placebo-controlled study. In the study with safinamide, on-time without troubling dyskinesia was increased by 1.4 h compared with 0.6 h with the placebo. Compared to a baseline of 9.06 (2.50) hours, safinamide reduced daily off-time by 1.6 h, while the placebo reduced off-time by 0.57 h (least-squares mean difference, 0.96 h; 95% CI, 0.56–1.37 h; *p* < 0.001) [20]. These findings were in agreement with the results of the 016 trial.

A recent systematic review evaluated the effectiveness and safety of safinamide as an add-on treatment to PD with fluctuations in motor and non-motor functions. PD patients with motor fluctuations showed significant improvements in on-time without troublesome dyskinesia and off-time UPDRS-III with safinamide at 100 mg daily. Similar results were observed for the 50-mg dose, but the quality of the evidence was lower. Patients taking 100 and 50 mg showed a significant though the slight reduction in UPDRS-II [24].

As a result of a randomized phase II/III study conducted by Hattori et al. [22], it was determined that safinamide was safe and effective in the treatment of PD patients on the basis of a mean daily ON-time and mean daily OFF-time. There were significant improvements in the UPDRS Part II total score (OFF phase), UPDRS Part III total score (ON phase), and UPDRS Part I. Compared with the placebo group, the change in the mean daily ON-time was 1.39 h in the safinamide 50 mg group and 1.66 h in the 100 mg group at week 24 (*p* = 0.0002). In addition, in the multiple regression analysis of a phase II/III study, both doses of safinamide significantly improved cardinal motor symptom scores (bradykinesia, rigidity, tremor, axial symptoms, and gait disturbances) [25].

The XINDI study, a phase III, randomized, double-blind, placebo-controlled, multicenter study (*n* = 307 PD patients), with a 2-week screening period and a 16-week treatment period, supported these findings. Between the safinamide and placebo groups, a significant change in the mean OFF time was 1.10 h at week 16 (*p* = 0.0001). In the safinamide group, this change was significantly greater starting at week 2, suggesting a rapid onset of drug effectiveness [23].

Unlike safinamide, rasagiline is indicated for treating PD as monotherapy or as adjuvant therapy. A two-phase study comparing rasagiline to the placebo (*n* = 404 patients with early PD) was conducted in the TEMPO trial. During phase 1, rasagiline was evaluated for its efficacy and safety in treating early Parkinson’s disease. At the beginning of the study, patients had mean UPDRS scores of 24–25. When rasagiline 1 mg and 2 mg were used for 26 weeks, impairment increased by 0.1, 0.7, and 3.9, respectively [26]. Phase 2 evaluated the early versus late onset progression of functional disability in PD patients. In the placebo group, patients have switched to rasagiline 2 mg. The scores increased to 3.01 after one year with 1 mg rasagiline, 1.97 after 2 mg rasagiline, and 4.17 after switching from placebo to rasagiline [27].

In the PRESTO study, rasagiline as add-on therapy with levodopa showed a significative reduction in the daily off-time compared with the placebo (rasagiline 0.5 mg: −0.49 h [*p* = 0.02 CI: −0.91–0.008]; rasagiline 1 mg: −0.96 h [*p* < 0.001 CI –1.36, –0.51] [26]. These results were supported by another pivotal study (LARGO study), where rasagiline (1 mg) also reduced the off periods as an add-on therapy with levodopa (–0.8 h [*p* = 0.0001, CI –1.18, –0.41] [28] (Table 2).

In the ADAGIO study, rasagiline was randomized into a double-blind, placebo-controlled study for patients with early Parkinson’s disease. The patients were randomized to receive rasagiline 1 mg/day for 18 months, rasagiline 2 mg/day for 18 months, and placebo for nine months, followed by rasagiline 1 mg/day for nine months. Compared to the placebo groups, rasagiline at 1 or 2 mg/day was associated with a slower rate of worsening in the active drug groups. The mean total UPDRS score decreased less when rasagiline 1 mg/day was started early than when it was started late over 18 months. Despite this, the groups that began rasagiline early did not differ from those who started late. Based on their findings, the investigators concluded that early treatment with rasagiline at a dose of 1 mg/day could be capable of modifying the disease. Early treatment with rasagiline at 2 mg/day, however, did not work [25].

Overall, the efficacy of safinamide and rasagiline on motor symptoms in PD has been well-established through various studies [20,21,26]. These drugs have been shown to not only improve motor function but also the activities of daily living and quality of life.

### 3.2. Evidence of Efficacy of Safinamide and Rasagiline on Non-Motor Symptoms

PD has traditionally been considered a motor disorder. However, it is widely acknowledged as a complex disorder with both neuropsychiatric and non-motor manifestations beyond motor symptomatology. Cognitive dysfunction, dementia, psychosis, hallucinations, mood disturbances, sleep disturbances, fatigue, autonomic dysfunction, gastrointestinal dysfunction, pain, and sensory disturbances are among these features. [29,30]. Both safinamide and rasagiline may help non-motor symptoms [31]. In particular, safinamide presented positive results on non-motor symptoms using the Non-Motor Symptoms Scale for Parkinson’s Disease (NMSS) [32,33,34] and rasagiline in the PS-23 non-motor scale [35].

An analysis of pivotal studies found that the safinamide group reported a significant reduction in concomitant paint treatments compared with the placebo group (23.6% reduction vs. placebo, *p* = 0.0421). Additionally, two out of three pain-related items in the “Bodily discomfort” domain of the Parkinson’s Disease Quality of Life questionnaire-39 (PDQ-39) significantly improved [36]. In line with these results, another prospective observational study showed significant improvements in the King’s Scale for Parkinson’s Disease [37]. In a more recent open-label prospective study, the non-motor symptoms scale (NMSS) total score was reduced by 38.5% (from 97.5 ± 43.7 in V1 to 59.9 ± 35.5 in V4; *p* < 0.0001). Through domains, the improvement of pain stands out (−43%; *p* < 0.0001) [33]. No studies with rasagiline and pain were found during the search.

Non-motor symptoms such as depression are common in people with PD. In the same pooled analysis, a significant improvement in the GRID Hamilton Rating Scale for Depression (GRID-HAM-D) was observed in the safinamide group after 24 months when compared with the placebo group (mean difference vs. placebo: −0.57; *p* = 0.0408). Accordingly, in two separate studies, the PDQ-39 “Emotional well-being” domain also scored better in the safinamide group [18,33]. Similar results in HAM-D have been observed in a more recent observational retrospective study of a smaller sample [38]. Rasagiline has also shown clinical improvements in depression symptoms as measured by the HAM-D scale [33,39]. Additionally, another study on patients with novo PD found lower depression and cognition scores in the rasagiline group compared to the placebo. Thus, the authors suggested that rasagiline combined with antidepressants may prevent the worsening of a variety of non-motor symptoms in patients with de novo PD [40].

An analysis of fluctuating PD patients treated with safinamide and rasagiline to assess sleep disturbances and daytime sleepiness was conducted using the Parkinson’s disease sleep scale 2 (PDSS2), Pittsburgh sleep quality index (PSQI), and Epworth sleepiness scale (ESS). A significant improvement in PDSS2 and ESS scores was found in safinamide patients (*n* = 46), as opposed to rasagiline patients (*n* = 15) [41]. Improvements in PSQI and ESS scores and reductions in the sleep/fatigue domain have been observed in recently published studies on safinamide [34]. In line with these results, an 18-week randomized, double-blind, placebo-controlled trial with rasagiline 1 mg/day as an additional therapy showed an improvement in the SCOPA daytime sleepiness score. The mean global improvement was −3.6 in the rasagiline group and −1.2 in the placebo group [33,42].

The effect of safinamide on rapid eye movement sleep behavior disorder (RBD) symptoms was described in an exploratory longitudinal cross-over study (*n* = 30 patients with PD and RBD) [43]. Participants received safinamide (50 mg/day) as an addition to or as an alternative to the usual antiparkinsonian therapy for a period of 3 months. After treatment with safinamide, 22 out of 30 patients reported a clear improvement in their symptoms, and 16 were completely free of clinical RBD symptoms. RBD symptoms were reported to have improved in eight patients. The mean UPDRS-II and III scores decreased after safinamide treatment, while the PDSS-2 score indicated improved motor symptoms. There is good tolerability and improvement in RBD symptoms for patients with Parkinson’s with safinamide [43].

Urinary problems are also common non-motor symptoms of PD, and it has been suggested that safinamide can have a possible benefit on urinary urgency, incontinence, or nocturia [40]. Rasagiline could also have positive effects on the bladder and in the urinary tract urodynamics [44].

Safinamide and rasagiline have been shown to be effective in the control of non-motor symptoms, such as cognitive dysfunction and dementia, psychosis and hallucinations, mood disorders, sleep disturbances, and fatigue [31]. The usage of NMSS and PS-23 scales is particularly useful for measuring the effects of both of these drugs [32,34]. A prospective, randomized, double-blind, exploratory study in Spain included 30 non-demented PD patients (80% of them having clinically significant apathy symptoms according to the Apathy Scale (AS)). The primary outcome was the mean change between the baseline and week 24 on the AS. A trend toward significance was observed in AS (ANOVA) due to a more marked decrease in the AS score with safinamide (−7.5 ± 6.9) than with placebo (−2.8 ± 5.7). Safinamide reached significance in the primary analysis, and a significant benefit was observed between weeks 12 and 24 compared to the placebo [45].

### 3.3. Switching Therapy from Rasagiline to Safinamide

Researchers have shown that switching from rasagiline to safinamide improves the wearing-off phenomenon, which is a recurrent, predictable worsening of parkinsonism symptoms at the end of the levodopa dose until the next dose. [9,18,42,46,47,48,49]. In this sense, preliminary reports suggest that switching from rasagiline to high-dose safinamide may benefit fluctuating PD patients [18,25,50].

The MAO-I was changed in an observational, retrospective multicenter study with 91 PD patients (17 on rasagiline and 4 on selegiline) because motor fluctuations and dysfunctional dyskinesia persisted. By reducing the daily off time and time ON with disabling dyskinesias, this switch was associated with significant improvements in motor scale scores (UPDRS III score in ON phase, and UDysRS item 9 score). Safinamide also showed a significant reduction in the total LEDD [47].

The switch to safinamide was also evaluated in another observational, retrospective study (*n* = 17 patients with PD). After taking levodopa plus rasagiline or levodopa plus dopamine agonists, or after the re-occurrence of previously corrected fluctuations, patients were switched to safinamide 100 mg. This switch produced a clinical benefit in 9 of 17 (52.9%) patients, significantly reducing wear-off and produced no adverse events (8).

A prospective clinical practice study assessed the effects of safinamide on motor fluctuations in 47 patients with PD. Ten of these patients were previously treated with rasagiline and were switched to safinamide following the washout period of two weeks, as recommended. The mild to moderate worsening of parkinsonism was observed in 2/10 patients who were on rasagiline before starting safinamide treatment, and subsequent moderate (CGI: 2) improvement in the wearing-off and parkinsonism was observed after three months with safinamide 100 mg/day. On the other hand, in 3/10 patients, the withdrawal of rasagiline was accompanied by a decrease in biphasic dyskinesias (*n* = 1) and generalized choreoathetosis (*n* = 2), which did not worsen and even improved more clearly when starting treatment with safinamide 100 mg/day (45).

In line with these results, a prospective observational real-life study was performed on 90 patients with PD who started on safinamide; 68.8% of them received combined therapy with a dopamine agonist, and 52.7% of them were treated with rasagiline previously. After six months, there was a statistically significant decrease in morning akinesia in 33.3% of patients, wearing off in 34.4% and MDS UPDRS-III scores. The authors concluded that safinamide was safe and could significantly improve motor fluctuations, motor symptoms, and the subjective perception of PD severity [51].

The clinical global impression scale was used to assess clinical improvements in more than 75% of patients with motor symptoms (bradykinesia, muscular rigidity, resting tremor, and gait) as well as non-motor symptoms (cognitive function, attention, sleep, neuropsychiatric and sensory disturbances) after starting safinamide treatment in a multicenter retrospective cohort study (54% of whom previously used rasagiline). It was noticed that patients with previous MAO-B inhibitors that switched to safinamide reported significant clinical benefits on motor and/or non-motor symptoms [48].

First-time MAO-B inhibitor users in combination with levodopa were examined retrospectively to update the evidence regarding add-on therapy for PD. Among the 4734 patients treated with MAO-B inhibitors combined with levodopa, 1059 were first-time users. In the rasagiline cohort, 18% of patients switched to another MAO-B inhibitor, followed by patients treated with selegiline (11.0%) and those with safinamide (4.3%). More than 70% of subjects discontinuing rasagiline switched to safinamide. According to the study, switching from selegiline to rasagiline improved motor behaviors, motor complications, mood, and sleep in 30 patients with PD [52].

Before starting safinamide, some neurologists recommend a washout period from MAO-B inhibitors. Some advocate for a shorter washout period; others begin safinamide immediately after stopping rasagiline. A study assessed the safety and tolerability of the immediate switch, without the two weeks washout period, from rasagiline to safinamide (50 mg and 100 mg) in 20 patients. After the switch, there was no significant change in blood pressure, and no cases of serotonin syndrome hypertension were reported or any other adverse events. During the study, the results confirmed that safinamide was well tolerated and safe by patients [18].

Taking the studies cited before into account, the change from rasagiline to safinamide in patients with motor complications is a safe and tolerable therapeutic opportunity to optimize antiparkinsonian treatment, especially before considering possible advanced or second-line therapies. In line with these, a Spanish consensus concluded: 1. The switch from rasagiline to safinamide associated with levodopa could improve the motor status and non-motor symptoms; 2. For patients with PD previously treated with rasagiline who wished to switch to safinamide, a 2-week washout period may be avoided, and safinamide could be started immediately, although evidence is scarce; 3. Starting dosages of 50 or 100 mg/day could be recommended for rasagiline-treated patients switching to safinamide [49].

### 3.4. Safety and Tolerability

Safinamide is a safe and well-tolerated medication with similar rates of treatment-emergent adverse events (AE) compared to a placebo. Among the most common AEs observed in pivotal trials of safinamide were scotomata, blurred vision, asthenia, pyrexia, dizziness, abdominal pain, nausea, low back pain, headache, insomnia, hypertension, worsening of PD symptoms, and urinary tract infections [19,20,21].

Safinamide safety data were collected in the SYNAPSES study: a multinational, multicenter, retrospective-prospective cohort trial. It included 1610 patients aged >75 years, with relevant comorbidities and psychiatric conditions, who were followed up for 12 months. During the observation, 45.8% of patients experienced adverse events, 27.7% had adverse drug reactions, and 9.2% had serious adverse events. The adverse events associated with safinamide were those already described. The majority were mild or moderate and were completely resolved. Moreover, clinically significant improvements were seen in the UPDRS motor score [53]. The SYNAPSES study has shown a good safety profile in patients over 75 years of age. Therefore, it is not necessary to modify the dose in elderly patients [53].

The use of safinamide as adjunctive therapy to levodopa can increase dopamine levels, which could lead to the emergence or worsening of dyskinesias. As part of the Study S01, a total of 0.8% of patients received safinamide 100 mg/day, 0.9% received 50 mg/day, and 2.3% were given a placebo; the SETTLE trial included a total of 1.8% receiving safinamide and 0.4% receiving a placebo [19,20]. Safinamide and the placebo had comparable incidences of new/worsening dyskinesias in the extension study [21]. It has been proposed that the levodopa dose may be reduced, and safinamide dose may be initiated or increased to 100 mg/day to prevent these emergences or worsening dyskinesias [49].

The most common adverse event from rasagiline is headache, but it has also been associated with the development of ecchymoses, dyspepsia, gastroenteritis, depression, falling, malaise, paresthesia, vertigo, arthralgia, arthritis, neck pain and conjunctivitis, rhinitis, and flu-like symptoms. Specifically, the TEMPO study, which is a randomized, double-blind, placebo-controlled, parallel-group design with 473 patients, indicates that adverse events were no more frequent than in the placebo group, and the most common ones were infections (16%) and headaches (12%) [29].

In a clinical review of safinamide in 1949, patients were treated with safinamide, with 1438 of these patients being treated with the highest recommended dose (100 mg) for at least 12 months [54]. In general, serious AEs were more common in patients on safinamide than on the placebo. Throughout all the studies on safinamide, there were 61 deaths. In the controlled studies, the incidence of deaths was similar between the safinamide and placebo treatment groups (about 2.2/100 patient-years) [54].

Compared with rasagiline, safinamide had a higher selectivity for inhibiting MAO-B, making it a safer option, especially for the treatment of serotoninergic syndrome and hypertension. In contrast to rasagiline, safinamide could be metabolized by CYP1A2, and plasma concentrations are not influenced by commonly used drugs that are metabolized by it (fluoroquinolones, verapamil, amiodarone, insulin, omeprazole, etc) [49,55].

### 3.5. Therapeutic Role

The Movement Disorder Society has included safinamide in a different class from selegiline and rasagiline because of its different mechanism of action: MAO-B/glutamate release inhibitor, even though they belong to the same therapeutic class of MAO inhibitors. [41]. It increases dopamine availability within the striatum and decreases glutamate release in hyperexcitable glutamatergic regions [20,22]

As the therapeutic arsenal for PD grows, clinical symptoms can be better managed, and complications tend to diminish. PD patients often have polypharmacy, which can cause adverse events and drug interactions [56]. There may be severe neuropsychiatric complications associated with dopamine agonists, including hallucinations and delirium, as well as behavioral problems. Antipsychotics, tricyclics, and tetracyclic antidepressants may increase the risk of severe cardiac arrhythmias when combined with amantadine. Anticholinergic effects may impair cognitive function, and hallucinations and psychosis are possible side effects [50,55]

Finally, although so far it is an off-label use, the role of safinamide as an early monotherapy has been explored in a recent pilot open-label study. Fifteen recently diagnosed PD patients were started on safinamide 50 mg/day in monotherapy. The study reported that at 3 months, 14 out of 15 patients improved their MDS-UPDRS Part III motor score by a mean of 1.4 ± 0.9. After 6 months, 12 patients were on 100 mg/day, and two patients were on 50 mg/day, and their motor symptoms score had decreased by a mean of 1.6 ± 0.9 with respect to the basal visit. Adverse events were noted in three patients (20%), one of them leading to discontinuation because of severe dizziness and nausea [57].

Patients with fluctuating disease and insufficient motor control were studied in a single-center retrospective observational study to see if higher doses of safinamide were effective and safe [58]. Participants receiving safinamide 100 mg daily switched to safinamide 150 mg or 200 mg daily. We found that the UPDRS IV total score significantly improved (7.8 vs. 6.2, *p* = 0.007), mainly due to the longer off-time (2.0 vs. 1.3, *p* = 0.013). Nine out of the eighteen patients that were included (50%) had a Clinical Global Impression of improvement (CGI 1–2) regarding the duration of dyskinesia (16.7%), its functional impact (22.2%), pain from dyskinesia (11.1%), and the duration of rest (44.4%). This pilot study suggests that increased safinamide >100 mg might benefit patients with fluctuating Parkinson’s disease, even with advanced therapies, without worsening dyskinesia significantly [58].

Even though future research is needed to confirm these findings, the present study showed that safinamide might exhibit positive effects on symptoms and, therefore, provide an option for monotherapy treatment in patients with a recent diagnosis of PD and mild disease. So far, there are no comparative studies between MAO-Bs in this setting, which highlights the need for future research. Currently, there is an observational study, the SUCCESS trial (NCT03994328; n = 1235), that aims to evaluate the effectiveness of safinamide, rasagiline, and other standards of care drugs when prescribed as an add-on to levodopa. A variety of factors have been considered in this study, such as quality of life, pain relief, change in antiparkinsonian treatment, the use of concurrent painkillers, compliance with PD treatment, hospitalizations, and the use of other healthcare resources [59]. These types of studies may help clinicians better position each MAO-B along the course of and disease for PD.

## 4. Conclusions

MAO-B inhibitors are a useful therapeutic option for Parkinson’s disease. Rasagiline has been shown to reduce the mean daily ‘off’ time and increase daily ‘on’ time without troublesome dyskinesias in fluctuating PD. Clinical trials have shown that rasagiline therapy is well tolerated and associated with a low incidence of cognitive and behavioral adverse events, making it a valuable treatment option for Parkinson’s disease.

Safinamide is an effective and safe long-term add-on drug in combination with levodopa in patients with PD. Its unique double mechanism of action positively provides further benefits to fluctuating PD patients and presents some extra benefits in both motor and non-motor symptoms compared to rasagiline.

Switching other MAO inhibitors to safinamide in patients with motor complications can improve wearing-off phenomena. Switching is a safe and tolerable therapeutic opportunity to optimize antiparkinsonian treatment, especially before considering possible advanced or second-line therapies. Together with its overall good safety profile, this may result in the valuable improvement of patients’ quality of life. Despite its promising effects in early PD, additional studies, and the inclusion of long-term and comparative studies, are needed to support current findings.

## Figures and Tables

**Figure 1 brainsci-13-00276-f001:**
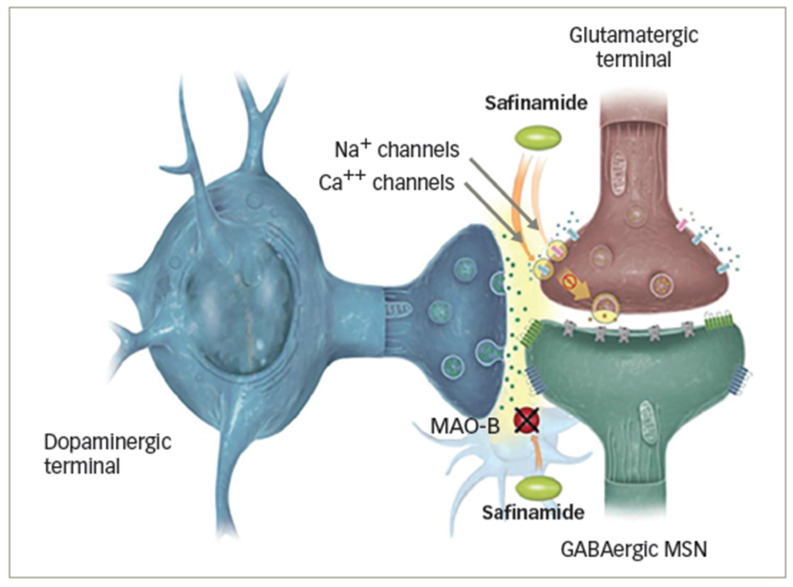
The novel mechanism of action from safinamide in Parkinson’s disease (PD) [14].

**Table 1 brainsci-13-00276-t001:** Characteristics of safinamide’s pivotal studies.

Study	Design	Participants of the Trial	Duration	Intervention Groups	Control Group
Study 016 Borgohain et al. [9]	Randomized, double-blind, placebo-controlled, parallel group.	669 adults (34–80 years) with PD in treatment with L-DOPA ± other drugs with motor fluctuations	24 weeks	Safinamide 50 mg Safinamide 100 mg	Placebo
Study Settle Schapira et al. [8]	Randomized, placebo-controlled clinical trial	549 adults (40–80 years) with PD in treatment with L-DOPA ± other drugs with motor fluctuations	24 weeks	Safinamide 50 mg Safinamide 100 mg	Placebo
Study 018 Borgohain et al. [15]	Randomized, double-blind, placebo-controlled, parallel group.	544 patients from study 016	18 months	Safinamide 50 mg Safinamide 100 mg	Placebo
Study by Hattori et al. [22]	Randomized, double-blind, placebo-controlled, parallel-group	406 patients, of whom 349 completed the study	24 weeks	Safinamide 50 mg Safinamide 100 mg	Placebo
Study XINDI Wei et al. [23]	Randomized, double-blind, placebo-controlled, parallel group.	307 patients with a diagnosis of idiopathic PD	18 weeks	Safinamide 50 mg Safinamide 100 mg	Placebo

PD: Parkinson’s disease.

**Table 2 brainsci-13-00276-t002:** Characteristics of rasagiline’s pivotal studies.

Study	Design	Population	Duration	Intervention Group	Control Group
Study TEMPO	Randomized, multicentric, double-blind, controlled, parallel-group trial	404 adults (>35 years) who had at least 2 cardinal signs of PD and whose disease severity was not greater than Hoehn and Yahrstage III	26 weeks	Rasagiline 1 mg Rasagiline 2 mg	Placebo
Study PRESTO	Randomized, multicentric, double-blind, controlled, parallel-group trial	572 with at least 2^1⁄2^ h of daily “off” (poor motor function) time, despite optimized treatment with other anti-PD medications.	26 weeks	Rasagiline 1 mg Rasagiline 0,5 mg (+levodopa)	Placebo
Study LARGO	Randomized, placebo-controlled, double-blind, double dummy, parallel-group trial.	687 patients with idiopathic PD, defined by the presence of at least two cardinal signs of the disease (resting tremor, bradykinesia, and rigidity) without any other known cause of parkinsonism, and a modified Hoehn and Yahr11 stage of less than 5 in the off-state.	18 weeks	Rasagiline 1 mg (+levodopa) Entacapone 200 mg (+levodopa)	Placebo

PD: Parkinson’s disease.

## Data Availability

Data available upon request.

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
