# Peer review of "Switching from Rasagiline to Safinamide as an Add-On Therapy Regimen in Patients with Levodopa: A Literature Review"

_brainsci, 2023, doi:10.3390/brainsci13020276_

Round 1

Reviewer 1 Report

The authors summarize evidence for the efficacy of rasagiline to safinamide as add-on therapy regimen in patients with levodopa to improve motor fluctuations. They list prior studies on the efficacy of safinamide and rasagiline on motor symptoms, efficacy of both agents on non-motor symptoms, switching the therapy from rasagiline to safinamide, safety and tolerability, and the therapeutic role of safinamide and rasagiline. The authors provided two interesting tables that summarize the characteristics of the pivotal studies on safinamide and rasagiline. From this discussion, the authors suggest that switching rasagiline to safinamide in patients with motor complications can improve wearing-off phenomena. The switch is a safe and tolerable particularly before considering potential advanced or second-line therapies.

While the summarized information is interesting and the review is well-written, the authors do not provide take-home messages at the end of the respective section for the reader.

Comments:    

1) The current title is confusing and does not clearly tell the aim of the current review. In order to attract the interest of more readers regarding the clinical relevance of the work, the title of the present review needs to be modified to describe its relevance in the clinical practice. I suggest being modified to:

“Switching from rasagiline to safinamide as an add-on therapy regimen in patients with levodopa: Attenuation of motor fluctuations”.

2) The provided sections read like narration for the evidence of discussed points without critical aspects/reflection points. At the end of each section, a take-home message is advised to be provided.

3) In the introduction section, the authors are advised to add a separate section for rasagiline and safinamide to address: the mechanism of action, adverse effects, interactions, and contraindications (if any).

4) The work lacks the future directions that will include limitations and what is next step to translate these findings to clinical settings

5) The authors are also advised to work on the current manuscript to minimize the similarity index (39%).

6) In line 208, the authors are advised to modify the title of section 3.3 “Switching therapy”. The authors may consider re-writing as “3.3. Switching therapy from rasagiline to safinamide”.

7) More recent 2022 references are advised to be added to the review. 

8) The manuscript needs to be checked by a native English speaker for grammar and typos. Examples are:

- In line 110, the authors state “Characteristics of the safinamide’s pivotal studies.”. Please, consider re-writing to “Characteristics of safinamide’s pivotal studies”.

- In line 150, the authors state “Characteristics of the rasagiline’s pivotal studies”. Please, consider re-writing to “Characteristics of rasagiline’s pivotal studies”.

Reviewer 2 Report

Use of safinamide and rasagiline on Parkinson’s disease: A Literature Review focused on the therapy switch 

With the expansion of aging population, there has been a sharp increase in both the global prevalence of Parkinson's disease and the related disability-adjusted life expectancy by 128% and 113% from 1990 to 2017, respectively. Currently, the global burden of Parkinson’s disease has more than doubled, which far outstrips Alzheimer's disease and other dementias. Parkinson’s disease has been known as the second-most common neurodegenerative disease, affecting approximately 1–2 out of 1000 people. Safinamide is a selective and reversible MAO-B inhibitor with a sodium channel inhibitory effect, and it also inhibits glutamate release in the basal ganglia. Results from international phase 3 studies And a Japanese phase 2/3 study have demonstrated the efficacy of safinamide (50 mg/day and 100 mg/day) in improving wearing-off and motor symptoms in patients with PD. This narrative review we explore the switch from rasagiline to safinamide in patients with motor complications, as a feasible and effective alternative to optimize antiparkinsonian treatment.

It is an important research topic. However, I have some suggestions and corrections to the article that are appended below.

Point 1: Lacks graphical abstract in the manuscript.

Point 2: Abstract is a good overview of the topic.

Point 3: There is a need for a separate section with the heading "Preclinical evidence for the efficacy of safinamide in PD" and to summarize them in tabular form.

Point 4: Add a figure to exhibit the multimodal profile of safinamide.

Point 5: : In section 3, there should be a schematic representation of the mechanism of safinamide in Parkinson’s disease (PD).

Point 6: In section “Safety and tolerability” Include some critical studies that were not discussed in this review.

Committee for Medicinal Products for Human Use (CHMP) Assessment Report

www.ema.europa.eu/contact

Tertiary Pharmacology Review

https://www.accessdata.fda.gov/drugsatfda_docs/nda/2017/207145Orig1s000PharmR.pdf

Summarize major findings on acute, sub-acute and chronic toxicity studies of safinamide in the tabular form.

Reviewer 3 Report

Review of a manuscript “Use of safinamide and rasagiline on Parkinson’s disease: A Literature Review focused on the therapy switch” by Pilar Sanchez Alonso and coauthors submitted to “Brain Sciences”

Parkinson’s disease is a severe neurodegenerative disease the second after Alzheimer’s disease bringing enormous suffering for the patients and their relatives and huge contribution and loss for the health care providers. The is no efficient medication modifying the course of this disorder.  The aim of the manuscript is to summarize the current knowledge on the therapeutic use of safinamide and rasagiline in PD patients, as well as to compile all the evidence on switching from rasagiline to safinamide. This is an important biomedical area and the information presented in this review wil be interesting to the readers of the “Biomedicines”.

The following corrections and additions should be made:

Abstract

Lines 15-16: ”Add-on therapeutic drugs, such as dopamine agonists and monoamine oxidase B (MAO-B) inhibitors, such as safinamide and rasagiline, may be preferable to continuously increasing levodopa dose to optimize motor control in PD.” This is an awkward sentence which can be corrected as follows:”

“Add-on therapeutic drugs, such as dopamine agonists and monoamine oxidase B (MAO-B) inhibitors, for example, safinamide and rasagiline may be a desirable addition to continuously increasing levodopa dose for optimization of motor control in PD.”

Introduction

Lines 31-32. After the sentence : ”PD is a complex, progressive and age-related disease characterized by the loss of dopaminergic neurons in the pars compacta of the substantia nigra and the accumulation of misfolded α-synuclein called Lewy bodies [1]” the authors should add a citation on a recent review on PD: ”Biomarkers in Parkinson’s Disease”. Chapter in a book. Peplow P.V., Martinez B., Gennarelli T.A. (eds) Neurodegenerative Diseases Biomarkers. 2022. Neuromethods, vol 173. pp 155-180. Humana, New York, NY. https://link.springer.com/protocol/10.1007/978-1-0716-1712-0_7

Results

Table 1. “Population” is not an appropriate term here, it can be replaced by “Patients” or “Participants of the trial”

Line 111. “PD: Parkinson disease.” Is not necessary here and may be deleted.

Lines  277-278: ”Safinamide is a safe and well-tolerated medication, with similar rates of treatment-emergent adverse events (AE) compared to placebo. Among the most common adverse events (AEs)

The authors should use abbreviation only once: ”Safinamide is a safe and well-tolerated medication, with similar rates of treatment- emergent adverse events (AEs) compared to placebo. Among the most common AEs…”

Line 318:” As the therapeutic arsenal for Parkinson's disease grows, clinical symptoms can be…” After the authors used abbreviation for Parkinson’s disease they should use it in abbreviated form only.

Conclusions.

Line 358. “Together with its overall good safety profile, this may result in a valuable improvement in patients' quality of life (QoL)” There is no need to give abbreviation in the last sentence of the manuscript which was not further used.

References

In references 3 and 30 the name of the same authors is given in different spelling: 3. Warren Olanow C, 30. Olanow CW. It should be consistent

Round 2

Reviewer 2 Report

·       Most of the suggestions have been incorporated by the authors in the revised manuscript. Therefore, no issue with considering it for publication.